# Different Sampling Frequencies to Calculate Collective Tactical Variables during Competition: A Case of an Official Female’s Soccer Match

**DOI:** 10.3390/s22124508

**Published:** 2022-06-14

**Authors:** Ibai Errekagorri, Julen Castellano, Asier Los Arcos, Markel Rico-González, José Pino-Ortega

**Affiliations:** 1Department of Physical Education and Sport, Faculty of Education and Sport, University of the Basque Country (UPV/EHU), Lasarte 71, 01007 Vitoria-Gasteiz, Spain; julen.castellano@ehu.eus (J.C.); asier.losarcos@ehu.eus (A.L.A.); 2Society, Sports and Physical Exercise Research Group (GIKAFIT), Department of Physical Education and Sport, Faculty of Education and Sport, University of the Basque Country (UPV/EHU), Lasarte 71, 01007 Vitoria-Gasteiz, Spain; 3Department of Didactics of Musical, Plastic and Corporal Expression, Faculty of Education, University of the Basque Country (UPV/EHU), 48940 Leioa, Spain; markeluniv@gmail.com; 4Department of Physical Activity and Sport, Faculty of Sport Science, University of Murcia, Argentina 19, 30720 Murcia, Spain; josepinoortega@um.es

**Keywords:** women’s football, team behaviour, spatial-positioning variables, data collection, data processing, electronic performance and tracking systems

## Abstract

The objective of the study was to assess the impact of the sampling frequency on the outcomes of collective tactical variables during an official women’s soccer match. To do this, the first half (lasting 46 min) of an official league match of a semi-professional soccer team belonging to the Women’s Second Division of Spain (Reto Iberdrola) was analysed. The collective variables recorded were classified into three main groups: point-related variable (i.e., change in geometrical centre position (cGCp)), distance-related variables (i.e., width, length, height, distance from the goalkeeper to the near defender and mean distance between players), and area-related variables (i.e., surface area). Each variable was measured using eight different sampling frequencies: data every 100 (10 Hz), 200 (5 Hz), 250 (4 Hz), 400 (2.5 Hz), 500 (2 Hz), 1000 (1 Hz), 2000 (0.5 Hz), and 4000 ms (0.25 Hz). With the exception of cGCp, the outcomes of the collective tactical variables did not vary depending on the sampling frequency used (*p* > 0.05; Effect Size < 0.001). The results suggest that a sampling frequency of 0.5 Hz would be sufficient to measure the collective tactical variables that assess distance and area during an official soccer match.

## 1. Introduction

Tracking systems (global navigation satellite systems (GNSS) or global positioning systems (GPS), local positioning systems (LPS), and semi-automatic video cameras (OPT)) allow, based on the recorded positioning data, either in geographic coordinates (latitude and longitude) or Cartesian (x and y axes), the analysis of kinematic variables (e.g., displacements, acceleration), as well as individual (e.g., heat maps) and collective (e.g., average positioning of the players) tactical variables of a team (distances between players and/or spaces covered by a group of players) [1,2,3]. It is true that these technologies have been used predominantly in applied research environments such as the domain of space-time analysis, focusing especially on describing the physical or conditional performance of soccer players [4]. Each time there are more and more investigations that try to address the description of tactical behaviour of soccer players using position data [1,5,6,7].

From a practical point of view, a soccer match must be understood on the basis of reciprocal relationships between the state of movement of the attack vs. defence dimensions of both teams in interaction [8]. Therefore, collective behaviour arises from the interaction between players who seek to self-organise and co-adapt to resolve emerging situations in the changing environment that playing entails [9]. In this sense, the space-time analysis of the positioning of the players allows us to identify the structure of the team and its evolution as the game progresses, being able to describe and explain the adaptive dynamics of a group of players (or the total) during a match [6,10,11]. Recently, the classification of space-time variables has been grouped into three geometric primitives based on the geometric perspective [12,13,14]: point (node) [14], line (distance) [13], and polygon (area) [12]. The node represents, at a single point, the average position of a player, several players, or the entire team (e.g., the geometric centre (GC)) [14]. Second, distance refers to the relationship between a point and an oscillator (e.g., player-goal, player-line) or between two oscillators (e.g., player-player, player-GC, GC-GC) [13]. Third, the area considered occupied space (i.e., the space occupied by a group of players), the explored space (i.e., the relative positioning of the team), and the dominant/influential space (i.e., the region to which the players can get to before any other player, depending on the Voronoi regions) of the team or of several players [12]. Taking into account that the analysis of these tactical variables is based on the numerical quantification of the spaces, one of the main differences found in the analysis of the results between the variables of each group is the difference in magnitude, which could influence some factors of the data collection process such as sampling frequency [15].

The sampling frequency, that is, the amount of data recorded per second measured in Hertz (Hz), determines the accuracy of the measurement of the spatial positioning of team-sports players [2,3]. Mainly, the problem arises because few data per second could lead to an incomplete, or even erroneous, analysis of the collective behaviour of the players, while an excessive amount of data per second could, initially, influence the quality of the signal due to sensor noise, and secondly, delay the analysis of the results [3,16,17]. For this last reason, a greater amount of data per unit of time, which corresponds to a higher sampling frequency, does not necessarily imply better data quality [2]. Some theories suggest that the sampling frequency must be at least twice as high as the highest frequency given by the signal itself, thus respecting the Nyquist theorem [17], although its application is complicated in sports, aggravating the need to search for new alternatives. However, to date, the relevance of the sampling frequency of positioning sensors to measure collective behaviour in team sports has hardly been studied [2]. A recently published systematic review [2] evaluated the use of different positioning systems and the sampling frequency applied by each tool to measure collective variables of spatial positioning in team sports. It is worth mentioning that most used sampling frequencies to measure the GC, the distance between two points, and the area ranged between 0.4 and 100 Hz, between 0.4 and 50 Hz, and between 1 and 30 Hz, respectively. The results of this review pointed out the lack of agreement that harbours the problem that it poses, giving rise to new investigations that look for a solution. As a consequence, a recent study [15] opened the line of research by evaluating the impact of four different sampling frequencies (1, 2, 4, and 10 Hz) on the outcomes of different tactical variables (point: change in the position of the geometric centre; line: mean distance between players; and polygon: total area) during a soccer match with 8 players per team (7 + goalkeeper vs. 7 + goalkeeper). The authors found that the sampling frequency affected the outcomes of the change in the position of the geometric centre and the distance between players. The researchers suggested that the study outcomes should be compared with caution since GC-related and distance variables were measured using different sampling frequencies. In addition, it is pointed out that a smaller amount of data could be used to measure area variables with large magnitudes. Since the magnitude of the collective tactical variables varies considerably according to the type of measurement, the type of scenario (training or match), the training task, and the competitive level of the players [15], the authors suggested the need to carry out more studies that evaluate the impact of different sampling frequencies considering different types of collective tactical variables during training sessions and/or competition matches, in order to arrive at a consensus on the right amount of efficient data to be used. To date, this is the most unique study has investigated the impact of the sampling frequency on the outcomes of collective tactical variables in soccer [15]. Nevertheless, the study did not carry out its analysis during an official competition match and only three variables (i.e., change in the position of the geometric centre, mean distance between players, and total area) and four sampling frequencies (i.e., 1, 2, 4, and 10 Hz) were included. Taking this into consideration, the objective of the present study was to evaluate the impact of different sampling frequencies in the measurement of collective tactical variables during an official women’s soccer match. The working hypothesis was that it would not be necessary to use a great quantity of data to calculate the outcomes of the different collective tactical variables during an official soccer match.

## 2. Materials and Methods

### 2.1. Participants

For the aim of this study, 11 semi-professional soccer players (age: 26.0 ± 5.0 years; height: 167.0 ± 4.0 cm; body mass: 61.1 ± 4.3 kg) belonging to the same team of the Second Women’s Division of Spain (Reto Iberdrola) were monitored during the first half (lasting 46 min) of an official league match of the 2020–2021 season. The analysed team usually trained during five 90 min sessions a week, plus a competition match each weekend. 

### 2.2. Variables

Taking into account previous studies [1,18], the collective tactical variables were classified into three main groups: point-related variables (i.e., change in geometrical centre position (cGCp)), distance-related variables (i.e., width, length, height, distance from the goalkeeper to the near defender (GkDef) and mean distance between players (DbP)), and area-related variables (i.e., surface area (SA)). Table 1 shows the definitions of all variables used in the study.

### 2.3. Procedures

#### 2.3.1. Data Collection

In order to ensure the strict description of the use of technology, a recently published protocol was followed [19]. Based on the points of information that should be provided when using Global Navigation Satellite System (GNSS), 17/19 were scored. To obtain position data, the players were monitored with WIMU PRO devices (RealTrack Systems, Almería, Spain) using the GNSS. Each WIMU PRO device was placed on a vertical position between the players’ shoulder blades, in a pocket of a specific chest vest (dimensions of the devices = 81 × 45 × 16 mm). The devices were activated 15 min before the warm-up of the match to avoid the so-called “technological lockout” [3,16]. The GNSS device used in this study can operate at 10 Hz and it is compatible with the Galileo and American satellite constellation, which seems to provide more precision [20]. For the analysis, the data were collected in an outdoor soccer field, without any possibility that infrastructures affected the data collection. During the game, a mean of 12 satellites were connected with each device. The value of DDOP was 0.95. This equipment and its measurements are valid and reliable using the GNSS for time-motion analysis in soccer (distance covered variable: accuracy = 0.69–6.05%, test-retest reliability = 1.47, inter-unit reliability = 0.25; mean velocity variable: accuracy = 0.18, intra-class correlation = 0.951, inter-unit reliability = 0.03) [21], and has been awarded with the FIFA Quality Performance certificate. Additionally, the agreement of the data on the collective tactical variables during an official soccer match between GPS and LPS sensors (that has a very acceptable precision to estimate the position of the players on the pitch [22]) has been tested with an intra-class correlation coefficient greater than 0.84 [23].

#### 2.3.2. Data Processing

In order to evaluate the impact of sampling frequency on the measurement of collective tactical variables, eight different sampling frequencies were used: data every 100 (10 Hz), 200 (5 Hz), 250 (4 Hz), 400 (2.5 Hz), 500 (2 Hz), 1000 (1 Hz), 2000 (0.5 Hz), and 4000 ms (0.25 Hz). The download of the records was carried out through the SPRO software (RealTrack Systems, Almería, Spain) after the end of the match. To calculate the tactical variables from the players’ positions on the pitch, the data were transformed into raw position data (latitude and longitude) using the software’s GIS mapping application, which allows for all kinds of geometric shapes such as polygons or circles with millimetre precision (geographic information system). Once the data were filtered through the software, they were imported into a Microsoft Excel spreadsheet (Microsoft Corporation, Washington, DC, USA) to configure a matrix.

### 2.4. Statistical Analysis

Descriptive statistics data from variables were presented using mean and standard deviation. Tests for normality (Kolmogorov–Smirnov) and equality of variances (Levene) were applied. One-way ANOVA analysis of variance for independent samples was used to test for differences in the variables between the eight sampling frequencies (i.e., data every 100 (10 Hz), 200 (5 Hz), 250 (4 Hz), 400 (2.5 Hz), 500 (2 Hz), 1000 (1 Hz), 2000 (0.5 Hz), and 4000 ms (0.25 Hz)). Significant results were then analysed using post hoc Tukey’s test, whereas Games–Howell’s post hoc test was applied when the variances were not homogeneous. The effect size (ES) was also calculated to determine meaningful differences between the sampling frequencies with magnitudes classified as [24]: trivial (<0.2), small (>0.2–0.6), moderate (>0.6–1.2), large (>1.2–2.0), and very large (>2.0–4.0). The level of significance was set at *p* < 0.05. The statistical analysis was conducted using the software JASP 0.16 (University of Amsterdam, Amsterdam, The Netherlands) and a customised Microsoft Excel spreadsheet (Microsoft Corporation, Washington, DC, USA) for Windows.

## 3. Results

Table 2 shows the values (mean and standard deviation) of the collective tactical variables of the first half of a match measured with different sampling frequencies. It should be noted that there were only significant differences (*p* < 0.05) in the cGCp variable between the sampling frequencies analysed. The lower the sampling frequency, the higher the value of the cGCp variable (0.25 > 0.5 > 1 > 2 > 2.5 > 4 > 5 > 10).

Table 3 shows the ES values between the different sampling frequencies analysed for the cGCp variable. In the rest of the variables, the magnitude of the effect was null (ES < 0.001).

## 4. Discussion

The objective of this study was to evaluate the impact of different sampling frequencies (i.e., data every 100 (10 Hz), 200 (5 Hz), 250 (4 Hz), 400 (2.5 Hz), 500 (2 Hz), 1000 (1 Hz), 2000 (0.5 Hz), and 4000 ms (0.25 Hz)) in the measurement of different collective tactical variables (i.e., cGCp, width, length, height, GkDef, DbP, and SA) during an official women’s soccer match. The main finding was that the sampling frequency did not influence the outcomes of the collective tactical variables (*p* > 0.05; ES < 0.001), except in those of the cGCp variable. The results suggest that a sampling frequency of 0.5 Hz would be sufficient to calculate the outcomes of the collective tactical variables during an official soccer match.

There are studies that have evaluated the accuracy of the electronic performance and tracking systems to record speed variables and those derived from it such as acceleration or deceleration [25], and suggest that low sampling frequencies (e.g., from 1 to 5 Hz) would not be precise enough to measure locomotor variables. The error seems to be reduced when the sampling frequency increases to 10 Hz [26], but from here the precision does not seem to improve. However, to the authors’ knowledge, there is only one study [15] that has evaluated the impact of sampling frequency on the measurement of collective tactical variables. Researchers should know the limit of Hz above which noise will affect the outcomes at higher frequencies, as in software-derived data, the deletion of poor data must ensure that important position data is not deleted [2]. For this reason, researchers record at higher frequencies, which are later reduced. The key would be to choose a balanced sample rate on the raw data (which is not affected by sensor noise, but leaves no data unrecorded), as well as choosing a balanced time interval that allows good performance on the raw data and treatment of the data in the software (e.g., minimum and sufficient data that do not cause loss of information or cause redundancy of information). This will avoid delays in reports on the behaviour of the players, allowing acceleration of the training prescription [16].

This study attempted to provide more information on the impact that sampling frequency has on the calculation of collective variables from positional data, assessing a greater range of recording possibilities at different frequencies than in the previous study [12]. This could be useful to compare to what extent the outcome of the collective behaviour variables commonly used in their analysis could be influenced, taking into account the magnitude of the variables of each of the main groups (e.g., point-related variables (i.e., cGCp), distance-related variables (i.e., width, length, height, GkDef, and DbP), and area-related variables (i.e., SA)).

Except for cGCp, sampling frequencies did not influence the outcomes of all tactical variables (*p* > 0.05; ES < 0.001). The results of the cGCp variable coincide with those of the study by Rico-González et al. [15]. The difference was substantial, between moderate and large during the soccer match (7 + goalkeeper vs. 7 + goalkeeper) [15] and between small and very large during the official women’s football match. The differences depending on the sampling frequency used may be due to the fact that the cGCp variable supposes the measurement of the difference in the location of two points and this will depend, above all, on the chosen time interval (e.g., more time plus distance between two midpoints).

In relation to the variables related to distance, some results of this study partially differed from those of Rico-González et al. [15], because the authors found significant differences with a magnitude that varied from small to moderate in the values of the DbP variable between the sampling frequency of 10 Hz and two others (1 and 4 Hz), while in this study, no significant differences were found, the magnitude being null (*p* > 0.05; ES < 0.001). The magnitude of the mean distances between players was lower during the 7 + goalkeeper vs. 7 + goalkeeper [15] compared with the official women’s soccer match. In this sense, both studies suggest that the influence of sampling frequency may be greater when measuring smaller distances (e.g., dyads). In the case of the official match, in a wide playing space, a sampling frequency of 0.5 Hz seems to be sufficient to measure the tactical distance variables. However, caution is necessary when analysing training tasks in a smaller space, such as small-sided games, where the magnitude of distances may be lower than in official matches, making more data per second necessary.

On the other hand, the results of the SA variable found in this study coincide with those of Rico-González et al. [15], the magnitude of the differences between the values in both works being trivial. Given that this area variable shows large magnitude values, it seems that the sampling frequency does not have a substantial impact on SA outcomes in medium and large game spaces. Therefore, a sampling frequency of 0.5 Hz also seems to be sufficient to measure this type of variable.

## 5. Conclusions

In brief, it seems that it would not be necessary to use a sample rate greater than 0.5 Hz to calculate the outcomes of the collective tactical variables related to distance (i.e., width, length, height, GkDef, and DbP) and area (i.e., SA) during an official soccer match. The results of this study can be useful to analyse the records without excluding relevant information, but without an excessive amount of data that will delay the submission of reports and their subsequent exercise prescription. Future studies should investigate the impact of the sampling frequency on the outcomes of other area variables with less magnitude (e.g., major range) with the aim of determining the optimal amount of data to use for the measurement of the set of tactical variables during competition. In addition, it would be interesting to include factors related to the analysed tasks (e.g., dimensions, number of participants, task constraints, etc.) as well as to the participants (e.g., gender, competitive level, etc.) that could influence the validity of the sampling rate. In this way, data processing would be more efficient for researchers and technicians, and comparison between studies would be possible.

## Figures and Tables

**Table 1 sensors-22-04508-t001:** Definitions of the collective tactical variables.

Groups	Variables	Definitions
Point	cGCp	Mean team change in the geometrical centre position, understood as the distance in metres between two consecutive measured points of the centroid as the midpoint of the polygon. To calculate this variable, goalkeeper activity was excluded.
Distance	Width	Mean team width, understood as the distance in metres between the two furthest-apart players across the width of the pitch. To calculate this variable, goalkeeper activity was excluded.
	Length	Mean team length, understood as the distance in metres between the two furthest-apart players along the length of the pitch. To calculate this variable, goalkeeper activity was excluded.
	Height	Mean team defence depth, understood as the distance in metres between the furthest back player and the goal she is defending. To calculate this variable, goalkeeper activity was excluded.
	GkDef	Mean distance in metres from the goalkeeper to the near defender.
	DbP	Mean team distance in metres between all the pitch players. To calculate this variable, goalkeeper activity was excluded.
Area	SA	Mean team surface area, understood as total square metres (m^2^) of a polygon described by players as its vertex point and calculated using the convex hull calculation. To calculate this variable, goalkeeper activity was excluded.

**Table 2 sensors-22-04508-t002:** Mean (M) and standard deviation (SD) of the collective tactical variables for each sampling frequency.

Sampling Frequencies (Hz)	Number of Data (*n*)	Variables													
		cGCp (m)		Width (m)		Length (m)		Height (m)		GkDef (m)		DbP (m)		SA (m^2^)	
		M	SD	M	SD	M	SD	M	SD	M	SD	M	SD	M	SD
10	27,600	0.12	0.13	39.81	11.08	32.64	7.40	34.13	13.23	19.63	6.95	22.96	11.67	952.54	409.15
5	13,800	0.24 ^a^	0.21	39.81	11.08	32.64	7.40	34.13	13.23	19.63	6.95	22.96	11.67	952.55	409.17
4	11,040	0.30 ^a,b^	0.24	39.81	11.08	32.63	7.40	34.13	13.23	19.63	6.95	22.96	11.67	952.50	409.03
2.5	6900	0.48 ^a,b,c^	0.37	39.81	11.08	32.64	7.40	34.13	13.23	19.63	6.95	22.96	11.67	952.55	409.22
2	5520	0.60 ^a,b,c,d^	0.46	39.81	11.08	32.64	7.40	34.13	13.23	19.63	6.95	22.96	11.67	952.51	409.22
1	2760	1.19 ^a,b,c,d,e^	0.88	39.81	11.08	32.64	7.40	34.13	13.23	19.63	6.95	22.96	11.67	952.47	409.17
0.5	1380	2.32 ^a,b,c,d,e,f^	1.71	39.80	11.09	32.64	7.40	34.13	13.24	19.63	6.95	22.96	11.67	952.43	409.56
0.25	690	4.32 ^a,b,c,d,e,f,g^	3.19	39.80	11.07	32.62	7.40	34.14	13.23	19.63	6.92	22.96	11.67	952.33	409.05

Note: cGCp is the mean team change in the geometrical centre position, Width is the mean team width, Length is the mean team length, Height is the mean team defence depth, GkDef is the mean distance from the goalkeeper to the near defender, DbP is the mean team distance between all the pitch players, and SA is the mean team surface area. ^a^ > 10, ^b^ > 5, ^c^ > 4, ^d^ > 2.5, ^e^ > 2, ^f^ > 1 and ^g^ > 0.5 for a significance level of *p* < 0.05.

**Table 3 sensors-22-04508-t003:** Effect size between the sampling frequencies (Hz) for the collective tactical variable cGCp.

Hz	10	5	4	2.5	2	1	0.5
**5**	0.7 (M)						
**4**	1.0 (M)	0.3 (S)					
**2.5**	1.4 (L)	0.8 (M)	0.6 (M)				
**2**	1.6 (L)	1.1 (M)	0.9 (M)	0.3 (S)			
**1**	2.1 (VL)	1.7 (L)	1.6 (L)	1.1 (M)	0.9 (M)		
**0.5**	2.4 (VL)	2.2 (VL)	2.1 (VL)	1.8 (L)	1.6 (L)	0.9 (M)	
**0.25**	2.5 (VL)	2.4 (VL)	2.3 (VL)	2.2 (VL)	2.0 (VL)	1.5 (L)	0.8 (M)

Note: S is small, M is moderate, L is large and VL is very large.

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
