# Peer review of "Different Sampling Frequencies to Calculate Collective Tactical Variables during Competition: A Case of an Official Female’s Soccer Match"

_sensors, 2022, doi:10.3390/s22124508_

Round 1

Reviewer 1 Report

The idea of the article is interesting, my recommendations are the following:
I recommend that the keyword women's football be included in the keywords.
I recommend that table 1 be moved to section 2.2.
Tables 2 and 3 should be moved to section 3. I also recommend that an interpretation of these results be made.
Section 4 recommends making new analogies of the results of this study with results from previous studies.
You are missing the Conclusions section
My recommendation is to reorganize the research of this study into sections mentioning clearly and in detail the targeted issues.
Lines 191-195 repeat the idea, I recommend rewriting or deleting.

Reviewer 2 Report

Different Sampling Frequencies to Calculate Collective Tactical

Variables During Competition: A Case of an Official Female’s

Soccer Match.

In this study, the authors elaborate on their recent findings of the effect of GPS sampling frequency (0.25-10 Hz) on different tactical variables in soccer in training, by presenting the analysis of half a match of a women’s team in the Spanish second division.

Introduction: based on the authors’ recent study [15] it should be possible to generate some specific hypotheses about what sampling frequency is the best

Results

At several places, an effect size of 0.0 is mentioned. Effect size and p-values are hardly ever 0.0 , I suggest reporting this as ES <0.001 .

More importantly (major point): Table 2 contains the main results. I think that the results are not very surprising, but maybe I do miss something crucial. By reporting the average values over 46 min of play I do not conceive how one would expect sampling frequency (within the current range) to affect the outcome measures. I would only expect differences when (rapid) changes are studied in these variables. Isn’t the change of the different variables (for instance when one of the teams makes a rapid counterattack) at least of similar relevance to the average values across the whole 46 min of play? The similar SDs among sampling frequencies suggest that the changes were also similar. However, GPS data are not very accurate in assessing rapid accelerations of players, thus when positions of players change rapidly over time,  the tactical variables must display differences when the changes within several seconds (e.g. during a counterattack) have to be taken into account. So this either isn’t of any relevance and/or the authors should explain why it is sufficient /justified to only present the mean values over 46 min of play and leave out the changes within these variables over time.

Along this line of reasoning, it seems not very surprising that the only variable which is based on a change (difference) of two consecutive measurements (cGCP) in time is highly dependent on the sampling rate. Yet the authors present this finding as if this wouldn’t be very relevant (‘With the exception of cGCp,’ it reads in the abstract as if this finding is a sort of side note). If players with higher acceleration capacity were studied (first division male players for example) the effect of sampling frequency on cGCP would even have been greater.

 In addition, in the future additional (new) tactical variables for which higher sampling rates may be necessary may become of interest.

Discussion

‘The results suggest that a sampling frequency of 2 Hz would be sufficient to

calculate the collective tactical variables during an official soccer match’’ How is this conclusion derived from the current findings? If there aren’t any differences between sampling frequencies one could also use 0.5 Hz which would reduce data size/handling time.

I do not see the point made about sampling noise. Indeed, when higher frequencies are used one may start to pick up higher frequency noise, but higher frequencies may be necessary to accurately assess tactical variables in which acceleration of players is important. Moreover, to remove noise we can filter the data (as long as the noise is of a different frequency the main relevant frequency in the data of interest).

There is quite a lot of self-referencing……

Round 2

Reviewer 1 Report

I have no other comments.

Author Response

Dear reviewer,

Thanks a lot for your review.

Best regards.

Reviewer 2 Report

The working hypothesis was that it would not be necessary to use a great quantity of data to calculate different collective tactical variables during an official soccer match.

What do the authors mean by this? Do you mean that it was expected that a low sample frequency would suffice to calculate the presented variables?

Author Response

Dear reviewer,

Thanks a lot for your review. We feel that your comments and suggestions have allowed us to improve our work substantially.

Taking into account your last comment about the hypothesis of our work, we have corrected the phrase. I think it will understand better now. 

All the changes included in the manuscript are marked up using the “Track Changes” function.

Best regards.

This manuscript is a resubmission of an earlier submission. The following is a list of the peer review reports and author responses from that submission.

Round 1

Reviewer 1 Report

To improve the presentation the main methods and formulas have to be presented.

References are mainly with Spanish authors, some authors have more than ten citation. I recommend to reduce some of them and to include others. For example - describing principles and methods of this research.

It will be better if some more information about the measurement system parameters and accuracy will be presented.

Reviewer 2 Report

  • Abstract need to rewrite and include the novelty and achievable contribution. Try to represent useful results to be achieved in the abstract to make it simple and clearer.
  • The novelty of the work is not clear in the introduction part. Kindly include all attempted novelty in the bullet points and explain in the detail.
  • State-of-the-art is very limited. Kindly enhance and include a summary table about the state-of-the-art, showing the comparison between available and proposed technology and highlighting the novelty and advantages part.
  • Kindly compare the obtained results with the available literature. Include a comparison table in the paper
  • Finally, Include an image of the proposed approach to understand the workflow of the proposed research work in the paper
  • Kindly include the mathematical modelling of the proposed approach

Reviewer 3 Report

Abstract

The abstract introduces the future reader adequately to what the article will be about.
It explains well what has been done, to whom, with what and the results obtained. It also includes the main conclusion reached by the authors.

There are some spelling mistakes that should be corrected, for example, in line 20 the word "rec-order" is truncated at the end, breaking a syllable, although we know that it is a direct application of the journal's template... But it is still a spelling mistake that I ask you to please correct throughout the text.

Introduction

The authors adequately approach with an updated review of the research topic, this allows to know what is the state of the art on the measurement accuracy of GPS technology in official matches of women football players.

Materials and Methods
2.1 Participants
Please include the Code of Ethics identification number that UPV/EHU has given you. 

2.2 Variables
Please include references for the classification of the groups of tactical variables. Thank you for this contribution.

2.3. Procedures 
2.3.1. Data collection
Ok, good work and good explanation

2.3.2. Data processing
At the end of the text you indicate that you do an analysis of the data, please inform about the software used. It is obvious that the analysis (2.4) was not done with Microsoft Excel.

2.4. Statistical analysis
Excellent

3. Results
This section focuses accurately and without unnecessary information on the aspects investigated. Good work

Discussion

Undoubtedly this is the best section of the article in which previous works are compared with those known in this work and it shows scientists and trainers that they do not need more than 2 Hz of frequency in this instrument to obtain excellent quality information.

Round 2

Reviewer 2 Report

  1. Several conventional approaches have been proposed and developed in this domain, and each has its specific limitations. Kindly include that limitation in tabular form. And also enhance the state-of-the-art related to this work as several AI, machine learning and data analytics approaches have been implemented. But the explanation of that approaches are missing. Kindly include the state-of-the-art along with their advantage and disadvantages in the form of a summary table. If possible, kindly create a new heading which can show the related work
  2. It is very hard to understand the work for a research point of view. Kindly include a proposed approach heading and explain in detail with the help of a figure which can explain the whole paper idea and work in the step-wise-step manner
  3. Developed approach implementation is missing. Kindly include the mathematical implementation in detail
  4. In the result and discussion section, more validations are required. Kindly compare the results with one more technique and as well as with existing literature.
